# Evaluation of the Performance of a Composite Profile at Elevated Temperatures Using Finite Element and Hybrid Artificial Intelligence Techniques

**DOI:** 10.3390/ma15041402

**Published:** 2022-02-14

**Authors:** Wangfei Ding, Abdullah Alharbi, Ahmad Almadhor, Payam Rahnamayiezekavat, Masoud Mohammadi, Maria Rashidi

**Affiliations:** 1Academy of Traffic and Municipal Engineering, Chongqing Jianzhu College, Chongqing 400072, China; dingwfjtyszgcx@cqjzc.edu.cn; 2Department of Information Technology, College of Computers and Information Technology, Taif University, P.O. Box 11099, Taif 21944, Saudi Arabia; amharbi@tu.edu.sa; 3Department of Computer Engineering and Networks, Jouf University, Sakaka 72388, Saudi Arabia; aaalmadhor@ju.edu.sa; 4School of Engineering, Design and Built Environment, Western Sydney University, Penrith, NSW 2747, Australia; 5Centre for Infrastructure Engineering, Western Sydney University, Penrith, NSW 2751, Australia; m.rashidi@westernsydney.edu.au

**Keywords:** channel shear connector, artificial intelligence, prediction, multi-layer perceptron, feature-selection, elevated temperatures

## Abstract

It is very important to keep structures and constructional elements in service during and after exposure to elevated temperatures. Investigation of the structural behaviour of different components and structures at elevated temperatures is an approach to manipulate the serviceability of the structures during heat exposure. Channel connectors are widely used shear connectors not only for their appealing mechanical properties but also for their workability and cost-effective nature. In this study, a finite element (FE) evaluation was performed on an authentic composite model, and the behaviour of the channel shear connector at elevated temperature was examined. Furthermore, a novel hybrid intelligence algorithm based on a feature-selection trait with the incorporation of particle swarm optimization (PSO) and multi-layer perceptron (MLP) algorithms has been developed to predict the slip response of the channel. The hybrid intelligence algorithm that uses artificial neural networks is performed on derived data from the FE study. Finally, the obtained numerical results are compared with extreme learning machine (ELM) and radial basis function (RBF) results. The MLP-PSO represented dramatically accurate results for slip value prediction at elevated temperatures. The results proved the active presence of the channels, especially to improve the stiffness and loading capacity of the composite beam. Although the height enhances the ductility, stiffness is significantly reduced at elevated temperatures. According to the results, temperature, failure load, the height of connector and concrete block strength are the key governing parameters for composite floor design against high temperatures.

## 1. Introduction

Since composite beams have great advantages such as considerable span length, small floor depth, and high stiffness, they are widely used in a variety of structures and buildings [1,2,3]. Shear connectors are typically employed in composite floor systems due to their superior shear performance in comparison with other structural elements. Different types of shear connectors have been proposed in which each one could address a significant issue of composite floor systems [4]. Generally, connectors are divided into two major groups including C-shaped and L-shaped connectors, while other cross-sections like V-shaped and I-shaped connectors are also used in different cases. One of the crucial problems of all shear connectors is their ability to retain shear strength at elevated temperatures [5].

Limited studies performed push-out tests with different loading patterns to evaluate slip and failure loads in the channel shear connectors. Channel shear connectors exhibited ductile performance; however, this behaviour was amplified in more extended channels. Nevertheless, the composite beams demonstrated brittle behaviour when channels were embedded in unconfined plain concrete. In contrast, when the channels were embedded in high-strength concrete (HSC) [6], the behaviour of the composite beam was ductile [7,8,9,10]. In addition, extended channel shear connectors represent better flexibility compared to lower channels. Bearing capacity increases linearly with length in a way that a C-shaped channel with 150 mm length has almost 60 percent higher load carrying capacity in comparison to a 100 mm channel. Moreover, when the C-shaped channel is embedded in high-strength concrete, failure modes are determined by concrete [11,12,13,14,15]. Although slip between the I-beam and the slab is inevitable, it could be considered small with an appropriate shear connector design. Thick channel connectors provide lower slip and higher load capacity [16]. Using engineering cementitious concrete (ECC) produced with synthetic fibres [17,18] along with channel connectors would increase both ductility and loading capacity, especially in reversed low-cycle loading [19,20,21,22].

Composite materials are formulated to possess the required properties and are employed as alternatives to natural and simple materials. Typically, composite materials are designed according to the specific application, and some of them are utilized for a variety of applications after successful testing [23,24,25,26,27]. Structural elements always have significant problems retaining their strength during heat exposure [8,28]. Many studies have taken variety of approaches to mitigate fire-induced damage to steel and concrete [29]. These studies are highly valued not only to increase safety of occupants during and after the fire, but also to decrease the refurbishment and retrofitting costs [30,31]. There are limited studies on the behaviour of shear connectors at elevated temperatures [32,33]. Combined with profiled slabs, shear studs have better performance due to steel profile coverage as a shield around the concrete against fire damage. Since lightweight concrete has better strength against fire, shear studs embedded in lightweight concrete show better ductility compared to studs embedded into normal concrete [6,32,34,35]. Three main types of failure have been observed during the experimental tests including shear connector fracture, concrete crushing and concrete shear plain failure. According to experimental studies, connectors will lose their strength in the case of fire but this deterioration can be changed in different situations [36,37,38,39,40]. However, few studies have investigated the behaviour of channel shear connectors at elevated temperatures. Thus, it is a subject that needs to be covered by more robust and comprehensive experimental studies. Reverse-channel connections enhance ductility at elevated temperature, but the stiffness decreases dramatically as the ultimate strength [41]. According to analytical studies, channel shear connectors working with HSC blocks can resist the deterioration induced by fire during the first 10 min of exposure. However, the failure may be an over-turning mode which does not correspond to failure mode at ambient temperature. Fire deteriorates the ductility of the connector [42]. In addition, C-shaped connectors can perform better than other shear connectors at elevated temperatures, which indicates high energy absorption [43,44,45,46]. According to previous studies, channel connectors have appealing features which make them suitable for steel–concrete composite structures. However, the main issue of the channel connectors is their performance at elevated temperatures. Therefore, finding an approach to cover the stiffness and ultimate capacity loss during fire exposure is an important goal.

As one of the supplementary approaches for empirical tests, the finite element method has always been used to verify the test results and extract additional data from the samples [47,48,49]. Furthermore, different programs have been produced based on FE principles, including ABAQUS and ANSYS, which are employed in various applications in engineering problems. Simulating composite materials has always been controversial among researchers. Although finding the best way to produce a reliable model is an issue, the FE-based programs have gone some way towards to addressing this demand. For example, the ABAQUS program has been successfully used for modelling the steel–concrete composite structures [50], and the results were reliable enough to be used in further studies. In another study, ABAQUS was utilized to simulate monotonic testing to verify experimental results and simultaneously provide new data from other samples [51].

Recently, using artificial intelligence (AI) techniques has become a common approach to address some of the technical problems in civil engineering, especially to predict and evaluate mechanical properties [52,53,54,55,56,57]. The nature of AI algorithms is based on learning and mocking, which makes them a favourable approach to avoid further experimental tests. AI techniques including machine learning and neural networks have been proposed and performed on a variety of engineering problems. Neural networks (NNs) are a common group of techniques that are employed to predict and analyse test results [58,59,60,61]. The development of different types of NNs has led to several algorithms including artificial neural networks (ANNs), adaptive neuro-fuzzy inference system (ANFIS) and multi-layer perceptron (MLP) [62,63,64,65].

The structural performance of shear connectors has been evaluated using NN algorithms. In a study, three well-known algorithms including Extreme Learning Machine (ELM), ANFIS and ANN were applied on the test results of shear strength from tilted angle connector samples and based on the results, all three algorithms produced competitive outcomes, while ELM performed slightly better than other algorithms [66]. The combination of ANN with PSO was developed in a study to predict the slip value of channel shear connectors embedded in normal and high strength concrete, where PSO showed a considerable role in improving the accuracy of the prediction [67]. Based on a review study, the ANN algorithm has been typically used to evaluate the shear strength of composite beams, whereas ANN was developed using optimization algorithms such as PSO, genetic algorithm (GA) and independent component analysis (ICA), while PSO represented better results compared to other techniques. According to review research, ANN and machine learning have been extensively employed to predict the performance of construction materials faced with fire, and the results indicated the ANN is a suitable technique in the prediction and evaluation of materials’ properties at elevated temperatures [68]. There are limited studies on the application of AI on the performance of shear connectors at elevated temperatures. The performance of angle shear connectors at elevated temperatures has been investigated using ANFIS and ELM combined with PSO and GA techniques, and the combination of ANFIS-PSO-GA demonstrated better outcomes. The combination of ANN and ELM with GA was used to predict the slip strength of C-shaped connectors at ambient temperatures of 550, 700 and 850 °C, and it was found that ANN and ELM are both capable of precise prediction of composite floor properties facing different heat stages. ELM, however, represented the lowest processing time [5]. MLP networks are another type of NN that have not been widely used in civil engineering problems, especially for evaluating the behaviour of materials at elevated temperatures. However, some studies have successfully developed MLP algorithms to predict the properties of structural elements. In a study, MLP was developed with PSO to predict the flexural strength of thin-walled sections. It was reported that MLP-PSO was able to accurately predict the test results. The MLP neural network is suitable for prediction, especially in problems with stochastic irregularities. An MLP–PSO combination was used in another study to predict the compressive strength of upright columns, and the combination of feature selection technique with NN method led to promising results [51].

There are limited studies in both experimental and numerical aspects of shear connectors at elevated temperatures. In this research, the AI techniques based on the feature-selection method were employed to find the governing parameters of channel shear connector design amongst the existed characteristics that affect the tensile strength of the steel–concrete composite floor system and to predict the shear behaviour in composite shear connector system. First, in order to obtain a proper database of relevant data, a finite element (FE) model was constructed and validated using ABAQUS software so that the models constructed were in good agreement with the test results. Push-out tests were also simulated using the ABAQUS program, and results were collected into datasets. Secondly, with respect to the reviewed studies, a hybrid AI algorithm was developed to predict the FE results. Accordingly, a feature selective (FS) algorithm was used in addition to a hybrid neural network including MLP and PSO algorithms to predict the slip strength of the channel connector at elevated temperatures and select the most crucial parameter in the design of composite floor system.

## 2. Experimental Program

In this part, an experimental procedure for the study of channel connectors at elevated temperature is briefly discussed, similar to the procedure used in [5]. The monotonic push-out tests were performed on channel shear connectors at elevated temperatures to investigate the strength properties of the composite system at high temperatures.

The test sample is a composite floor system that included an IPE270 profile, a channel as a connector that has been welded to each profile flange by fillet weld and two concrete blocks with dimensions 150 mm × 250 mm × 400 mm placed on both sides of the IPE270 profile (Figure 1). In order to prevent the cracking of concrete, a transverse closed rectangular reinforcement with a diameter of 10 mm per concrete block was provided. The reinforcement applied in rectangular closed stirrups in two upper and lower rows was kept in two rows of vertical reinforcement.

A typical load–slip curve is depicted in Figure 2 [69]. The knowledge about the stiffness of composite beam has been primarily used in the equation of the partial interaction theory of composite steel–concrete beams. Since there is no linear relationship between load and slip, a few mathematical expressions are needed to explain the shear connector load–slip relationship throughout the curves. Therefore, it is difficult to find a typical regression formula for the shear connector stiffness. Based on the temperature corresponding to the taken time, standard provisions have identified the typical fire curve for performing elevated temperature tests and the three most common standard fire curves are shown in Figure 3.

## 3. Finite Element (FE) Modelling

The ABAQUS/CAE v.14.1 (2014, Simulia, Providence, RI, USA) was employed to model the presented test specimen(s). The Finite Element (FE) models were adjusted to replicate the empirical tests [70]. To assure that the FE model was able to verify the real specimen(s), a range of element types and mesh matrices were carried out as trial-and-error procedures to find a sustainable structured mesh [71,72]. Figure 4 shows a 3D view of the model and the inner cut to demonstrate the position and details of the shear connector and transverse rectangular reinforcements.

### 3.1. Model Prepration

The selected interaction for different sections was surface-to-surface mode, and different constraints were defined for different parts of the frame [73]. Each channel connector was assumed to be a fixed member and defined as a tie section on the top of the flange. In addition, the diagonal bracing connections were simulated as couplings. The contact properties were designated as tangential contacts with a 0.3 friction coefficient. The lateral loads were applied to the main reference point which was placed at the exterior edge of the web. Moreover, the monotonic loading was acquired using the time coefficient description. A combination of the quadrilateral four-noded shell (S4R) and linear triangular (S3) elements was conducted to assimilate the composite sections (Figure 5). Table 1 presents the employed geometrics of the channels which have been simulated in the ABAQUS. Other properties and features used were as discussed in the experimental section.

To simulate the effect of fire on the model, Eurocode2 and Eurocode3 British Standards Institution provisions were employed for concrete and steel, respectively. Figure 6, Figure 7, Figure 8 and Figure 9 show the used standard curves for both concrete and steel in the FE model.

### 3.2. Model Authentication

Figure 10 shows the step-by-step sequence of the typical shear connector deformation at the ambient temperature. The exhibited deformations could be verified with the real test specimens.

Figure 11 illustrates a schematic view of the typical channel at 700 °C temperature, where the bottom of the connector has failed (the area in red). In addition, thinning occurred.

The governing failure mode was a ductile mechanism due to the connector failure at the bottom of the channel web while the concrete block was still serviceable. To demonstrate the authenticity of the ABAQUS results, Figure 12 shows two comparative curves for a better understanding of the accuracy and validation of the FE results. In these curves the models were able to acceptably simulate the load–slip curves for different heats and shear connectors.

### 3.3. Finite Element Results

Finally, the results of the FE models that employed both C7530 and C10050 shear connectors are presented in Table 2, and Figure 13 demonstrates the obtained load–slip curves for the aforementioned models.

Based on the obtained results, The C7550 and C10050 models indicated the most stiffened behaviour at ambient heat, and the C10030 model presented the most stiffened state at 850 °C. Both C10030 and C10050 models showed a ductile performance at elevated temperatures. According to Table 2, using the longer width increased the stiffness of the specimens at ambient temperature. On the other hand, employing longer height improved the ductility especially at 700 °C and 850 °C. The better stiffness at the ambient temperature could be related to the embedded length of the channel which could be increased by enlarging the width. It could also be due to the strength properties of the high-strength concrete block presenting a better bearing capacity. High ductility of the specimens at elevated temperatures could be related to the length of the channel connectors, where increasing the length increased the slip value, while failure load decreases due to fatigue in the channels. Using HSC blocks might be effective as a cover for retaining the rigid state of the steel at elevated temperatures, but this effect could not be considered for safety remarks [32].

## 4. Statical Data

The applied data in this research were derived from finite element results that eventually formed a database with 1010 rows of inputs. This database has seven inputs and one target output. The summary of this data is shown in Table 3.

## 5. Artificial Intelligence Prediction

In this paper, a combination of MLP with PSO algorithm based on the random production of the initial population is utilized. PSO is a universal method of minimization that can be employed to deal with problems whose answer is a point or surface in n-dimensional space. In order to identify the most influential input, instead of traditional methods, the feature selection technique is utilized, which is the best way to identify the most influential features of a problem.

### 5.1. Algorithm Methodology

#### 5.1.1. Multi-Layer Perceptron (MLP)

MLP network is a feed-forward algorithm that can be utilized as a powerful hyper-surface reconstruction tool that is able to successfully map a set of multi-dimensional input data $\left({\underline{x}}_{i};\;i=1,\ldots ,N\right)$ onto a set of appropriate multi-dimensional outputs $\left({\underline{y}}_{i};\;i=1,\ldots ,N\right)$. The MLP configuration has been extensively used in static regression applications, and it consists of one input layer, one or more hidden layer(s) and one output layer. The MLP network uses a supervised learning technique called backpropagation for training the network [74].

Figure 14 shows a schematic of the MLP neuron compositions and Figure 15 shows a schematic configuration of the single-layer MLP, which is used in the analysis of the current study.

#### 5.1.2. Particle Swarm Optimization (PSO)

In the PSO algorithm, particles are the building blocks of the population, and they work together to obtain the optimum approach to the target [51,62,67]. For this reason, it is called swarm intelligence. The most important feature of any particle is its position, and the critical issue is what indicator or target the particle offers and how fast it moves. In this study, the PSO algorithm is employed along with MLP as a unique intelligence algorithm. The goal of the PSO algorithm is to find the optimal objective function. A flowchart of the PSO algorithm is illustrated in Figure 16.

#### 5.1.3. Feature Selection (FS) Technique

FS method is preferred in cases where the readability and interpretation of the subjects are important because the discounted values are preserved as the main features in the reduced space. This method of dimensionality leads to the creation of a quality database without deleting helpful information [75]. The feature selection process is divided into four parts including production method, performance evaluation, stop criteria and validation method, which are shown in Figure 17.

In this study, some of the significant features of the channel shear connectors and the concrete blocks are identified through one or more conversions on the input features. Once mapping points from one ample space to another in a smaller space happens, many points may overlap. Feature extraction helps to find new dimensions that have minimal overlap [74]. Given this technique, we can safely say that there is no better combination of inputs than this technique provided. If we were to manually examine all possible modes, the 127 input combination modes would be impossible to verify. If we tried all the possible scenarios with only seven performances, we accelerated the impact of the inputs by about 18 times.

#### 5.1.4. MLP-PSO-FS (MPF) Technique

This study deployed a combination of MLP, PSO and FS to develop a hybrid algorithm called MPF for predicting the shear performance of a composite structure consisting of channel shear connectors at elevated temperatures. Figure 18 shows a sequential combination diagram of particle swarm optimization-feature selection (PSO-FS) and multi-layer perceptron (MLP). In PSO, congestion generally begins with a set of random solutions, with each one called a particle. The particle swarm moves in complex space. A function (f) is evaluated at each step by input. In the global version of the PSO, pi represents the most appropriate point in the entire population. A new velocity is obtained for each *i* particle in each iteration according to the best individual neighbourhood positions. The new speed can be obtained as follows:(1)$$v_{i}\left(t+1\right)=\left[wv_{i}\left(t\right)+c_{1}\emptyset_{1}\left(p_{i}\left(t\right)-x_{i}\left(t\right)\right)+c_{2}\emptyset_{2}\left(p_{i}\left(t\right)-x_{i}\left(t\right)\right)\right]$$

When the speed exceeds the specified limit, it will be reset to its proper range. Depending on the speed, each particle changes its position according to the following equation:(2)$$s_{i}\left(t+1\right)=s_{i}\left(t\right)+v_{i}\left(t+1\right)\;$$
where $\;s_{i}$ = particle’s position;$\;v_{i}$ = particle’s velocity;$\;p_{i}$ = most the appropriate position; w = inertia weight;$\;c_{1}$ and $c_{2}$ = acceleration coefficients; and $\emptyset_{1}$ and $\emptyset_{2}$ = uniformly distributed random vectors in [0, 1].

#### 5.1.5. Extreme Learning Machine (ELM)

Huang et al. [76] suggested ELM as an AI tool for single-layer feed-forward neural network (SLFN) architecture. In ELM, the weights of SLFN inputs are obtained randomly, while the output weights are analytically defined. The most remarkable advantage of the ELM algorithm is its speed in finding the weights of the network; in addition, it can determine all the network factors and prevent unnecessary interference of humans. Unlike other AI tools, ELM is a new tool, suitable for finding the weights of SLFN. Due to its benefits, ELM could gain popularity and workability. The three steps involved in ELM development are: (1) one SLFN is constructed, (2) the weights and biases of the network are randomly selected, and (3) the output weights are calculated by inverting the hidden layer output matrix.

#### 5.1.6. Radial Basis Function (RBF) Neural Network

In RBF, the input function (*f*_(*x*)_) is estimated by a set of D-dimensional radial activation functions. Figure 19 indicates the typical architecture of one network with an input layer of D neurons, the output layer of P neurons, a hidden layer of M neurons, biases at each output neuron and adjustable weights between the hidden and output layers. Regarding a set of N data points in a multidimensional space, the main goal of interpolation is to find a function in which every D-dimensional input feature vector (*x_n_* = {: *i* = 1…, *D*}) has a corresponding P-dimensional target output vector (*f_n_* = {: *k* = 1…, *P*}). The approximation function *f*_(*x*)_ might be explained as a linear combination of radial basis functions in which the output of network kth consists of the sums of weighted hidden layer neurons plus the bias when the system is represented by nth input vector as:(3)$$\hat{f_{k}}\left(X^{n}\right)=\overset{M}{\underset{j=1}{\sum }}w_{kj}h_{j}\left(X^{n}\right)+w_{k0},\;\;\;\;\;\;\;\;\;\;\left(k=1,2,\;\ldots ,P\right)$$
where $w_{kj}$ is the corresponding weight to *j*th basis function and *k*th output, $h_{j}\left(X^{n}\right)\;$ is the output from *j*th hidden neuron for the input vector $\left(x^{n}\right)$, and $w_{k0}$ is a bias term at *k*th output neuron.

## 6. Performance Evaluation

Three objective criteria, including correlation coefficient (*R*^2^), Pearson’s correlation coefficient (*r*) and root mean square error (*RMSE*) were used to evaluate the accuracy of the results and the reliability of the proposed neural network. *RMSE* is the most common criteria used to measure the accuracy of continuous variables with a quadratic scoring rule that also measures the average error rate. This square root is the average square difference between prediction and actual observation. In the case of r, a higher value, up to 1, represents a suitable fit between measured and predicted values, while a negative value shows that the model’s performance is worse than the average of the developed models. These statistical indicators can be characterized as follows:(4)$$\;RMSE=\sqrt{\frac{1}{M}\overset{M}{\underset{i=1}{\sum }}{\left(P_{i}-O_{i}\right)}^{2}}$$
(5)$$\;r=\frac{M(\sum_{i=1}^{M}O_{i}.P_{i})-(\sum_{i=1}^{M}P_{i})\cdot (\sum_{i=1}^{M}O_{i})}{\sqrt{\left(M\sum_{i=1}^{M}O_{i}^{2}-{\left(\sum_{i=1}^{M}\bar{O_{i}}\right)}^{2})\cdot (M\sum_{i=1}^{M}P_{i}^{2}-{\left(\sum_{i=1}^{M}\bar{P_{i}}\right)}^{2}\right)}\;}$$
(6)$$R^{2}=\frac{\sum_{i=1}^{M}\left(O_{i}-\bar{O_{i}}\right)\cdot \left(P_{i}-\bar{P_{i}}\right)}{\sqrt{\sum_{i=1}^{M}{\left(O_{i}-\bar{O_{i}}\right)}^{2}\sum_{i=1}^{M}{\left(P_{i}-\bar{P_{i}}\right)}^{2}}\;}\;$$
where $\;P_{i}=\mathrm{predicted}\;\mathrm{varaible}$, $O_{i}=\mathrm{observed}\;\mathrm{variable}$ and *M* = number of considered data.

Considering the number of data and avoiding overfitting, 70% of the inputs were randomly devoted to the training phase of the models, and the remaining 30% were assigned to the testing phase. All the codes were developed in the MATLAB environment, and available functions of the MATLAB v2019c software (2019, Mathworks, Natick, MA, USA) were used in the developing process.

## 7. Model Development

For the first time, this study uses the FS method to integrate with the MLP-PSO as a hybrid neural network (MPF) to predict the performance of the specific channel shear connector at elevated temperatures. Each algorithm has its own parameters that can be approximated by changing them. To achieve the desired results, a large number of implementations of neural networks with different configurations are taken, and the best settings are obtained for all algorithms and neural networks, which are presented in Table 4, Table 5, Table 6 and Table 7.

## 8. Results and Discussion

A database may contain a lot of input data but not all inputs are suitable for use in the neural network. Some have no effect on the output prediction, and some may cause network deviation. Therefore, with a large number of inputs, finding the best combination is very time-consuming. The number of possible combinations for k members of a set of n members is equal to (1 (n ≅ k)). For example, in this research, the number of inputs is seven, so the number of combinations of four inputs is 35 states and the total number of possible states is 127. It is evident that the implementation of the neural network and its results for this number of iterations is impossible given the different combinations of neural network settings, so the only way to select different input states and settings is based on past experiences and initial assumptions. Therefore, using the FS method is inevitable, and by running the FS method on our inputs, only seven runs of all inputs will now be checked, and the best combination will be identified. We start with the adjustment of the MPF neural network and try to find the best population with a constant number of repeats equal to 40. The population was identified equal to 125 and the corresponding results are presented in Figure 20 and Table 8 in the test phase.

After finding the best population, the calculations are performed again to find the optimal number of iterations. As shown in Figure 21 and Table 9, the number of optimal repeats in the test phase is 45.

After finding the best parameters for the neural network, it is time to find the optimal input combination through the feature selection technique. As mentioned earlier, in this technique, there is no need to test all possible combinations, and only each set of k members should be tested once. For example, if we want to determine the best combination of inputs with four members, we only run the network once and set nf to four, then four inputs are selected, which have the most effect on the response. Table 10 and Figure 22 specify the best value of k.

According to the results of Table 10, all the answers are accurate enough and reliable, which proves the validity of both experimental and FE results in datasets. Table 10 presents the results of the MPF algorithm based on the employed precise criteria, where model 4 was selected as the best prediction with impressive evaluation parameters including *RMSE* = 13.072, *r* = 0.972 and *R*^2^ = 0.945 and *RMSE* = 13.533, 0.970 and *R*^2^ = 0.941 for test and train phases, respectively. The small differences between test and train phase results for model 4 indicates that the MPF algorithm accurately predicted the behaviour of the composite floor system at elevated temperatures. In addition, models 2 and 5 were the next accurate predictions among other results. Furthermore, Table 11 indicates the most significant input combinations that the FS technique has chosen, where the load and temperature are the most important parameters for prediction of the slip value. However, four parameters, including load, temperature, compressive strength and channel height are marked as the most effective composition for prediction of the slip. Therefore, in order to enhance the shear strength and the compressive strength, the connector’s height should be constant in further designs, and other parameters could also be held constant or considered as lower priority. To limit the slip value, it should be focused on concrete design and finding the best configuration for channel connectors. The best compositions for predicting slip value based on the FS method were tabulated. Figure 23 and Figure 24 reveal the regression diagrams of the aforementioned input models, where four inputs show the most accurate prediction of the slip value. Figure 25 indicates the tolerance diagram of the four input models, in which both test and train phases have a slight difference between measured and experimental values.

After obtaining the best combination for the MPF neural network, it was necessary to check with one or two other neural networks whether the innovative hybrid network is working properly. In this study, RBF and ELM were employed to challenge the MPF results. Given that the best input parameters were obtained by the feature selection technique, trying different combinations at this stage was avoided and two combinations with four and seven inputs are used as per Table 11. In addition, the neural network settings were as per Table 6. Figure 26 demonstrates the regression charts of the ELM and RBF, in which the answers for both seven- and four-input models are accurate, indicating the efficiency of using the FS method in this paper and justifying the application of FS in other, similar studies. Figure 27 and Figure 28 present the tolerance diagram with respect to the train and test results of the RBF and ELM, respectively.

As shown in Table 12 and Figure 29, the best performance parameters for the MPF are *RMSE* = 13.072, *r* = 0.972 and *R*^2^ = 0.945, which are indicated as FS-4. As for the ELM, *RMSE* = 13.286, *r* = 0.969 and *R*^2^ = 0.938. In addition, for RBF in the test phase, *RMSE* = 13.884, *r* = 0.969 and *R*^2^ = 0.939 in the test phase. The best result for *RMSE* is the lowest value, while for *r* and *R*^2^, the best result is 1; therefore, the values closer to 1 are better results. Considering all the conditions stated above, it is clear that the MPF algorithm performs better than the other two algorithms.

## 9. Conclusions

The present study aimed to predict the strength properties of a specific C-shaped shear connector at different heat stages. To this end, a new AI technique was developed to evaluate the results. Firstly, finite element (FE) analysis was applied by the ABAQUS program to model the composite floor system and simulate push-out tests at elevated temperatures. Then the results were collected, and a validated database was established. Secondly, a well-known neural network algorithm called multi-layer perceptron (MLP) was developed with the particle swarm optimization (PSO) technique to present a novel hybrid algorithm for evaluating the FE data. In addition, the feature selection (FS) method was used to avoid trying all possible input modes and wasting time while providing the best possible input combination, which may be neglected in other ways. Using the FS method facilitated the prediction process for the hybrid algorithm. In this regard, FE modelling was discussed by explaining the mechanical and geometrical properties of the composite model along with the validation procedure, and the results were presented. Next, the MLP-PSO-FS (MPF) methodology was developed. Then the AI results were interpreted and to verify the MPF results they were compared with the results of two well-known algorithms, namely ELM and RBF. The dataset used consisted of 1010 rows of laboratory data including seven inputs, namely length (mm), fc (N/mm^2^), channel-thickness (mm), profile-thickness (mm), height (mm), temperature (°C), and failure load (kN), while the slip (mm) was considered as the output. The MPF obtained the best results using the feature selection technique, which was followed by a description of each neural network. The most important results can be summarized as follows:Based on FE results, using longer channels could increase the ductility of the composite system at lower heats; however, at elevated temperatures, the stiffness of the composite system experiences a noticeable loss.According to FS technique results, the failure load and temperature are the most effective inputs that can help to accurately predict slip value without using other inputs. Furthermore, concrete compressive strength and connector height are the two key parameters for a sustainable design of a composite floor system at elevated temperatures.The combination of an MLP neural network with the PSO optimization algorithm based on a random population achieved the best results with excellent accuracy. The result of the MPF algorithm on the model with a combination of four inputs was the most precise prediction with *RMSE* = 13.072, *r* = 0.972 and *R*^2^ = 0.945.ELM and RBF were also applied on the main models (four and seven inputs) to predict slip value. Both had better performance on seven-input models with *RMSE* = 13.286, *r* = 0.969 and *R*^2^ = 0.938 for ELM, and *RMSE* = 13.884, *r* = 0.969 and *R*^2^ = 0.939 for RBF.

Finally, despite the rankings of these three algorithms, it should be noted that the results of all three networks performed very well in the prediction of slip value at elevated temperatures. However, combining PSO with MLP provided the best results. In the RBF and ELM models, the best results were obtained with seven inputs but, similar to the MLP-PSO, the results with four inputs were very acceptable. Performing cyclic loading on channel connectors could be considered as further study. In addition, other geometrical shapes and sizes could be developed to investigate their performance as stiffeners.

## Figures and Tables

**Figure 1 materials-15-01402-f001:**
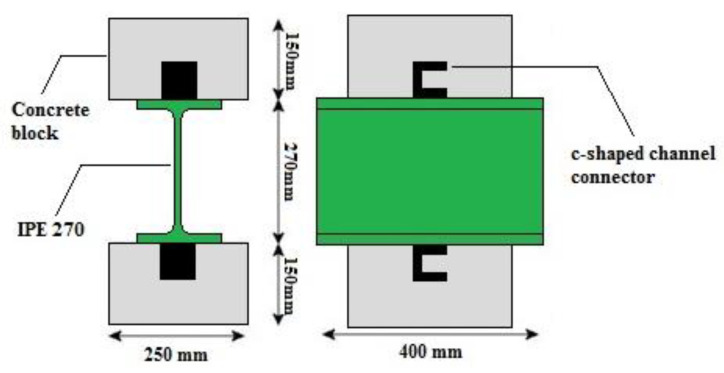
Schematic view of the test specimen.

**Figure 2 materials-15-01402-f002:**
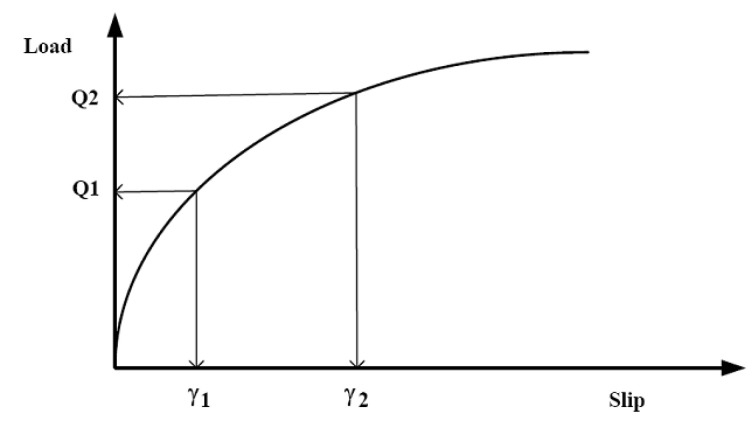
Typical load-slip relationship from the standard test.

**Figure 3 materials-15-01402-f003:**
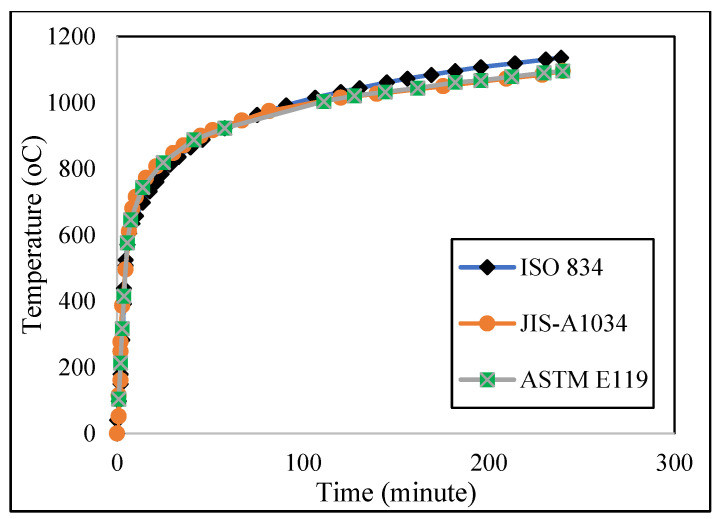
Standard fire curves.

**Figure 4 materials-15-01402-f004:**
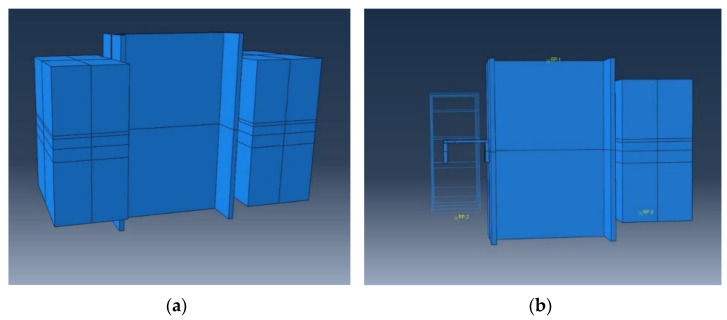
The sketched 3D model in ABAQUS: (**a**) exterior view and (**b**) inner cut.

**Figure 5 materials-15-01402-f005:**
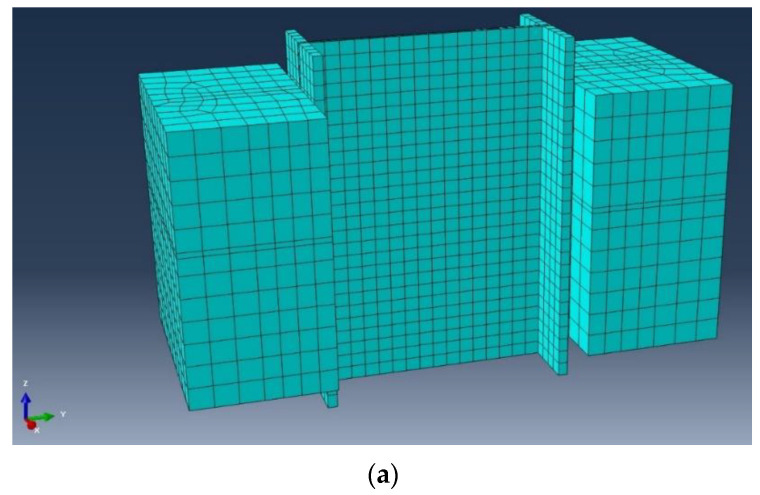
The modelled mesh and element lattices: (**a**) main model, (**b**) channel section and (**c**) IPE beam.

**Figure 6 materials-15-01402-f006:**
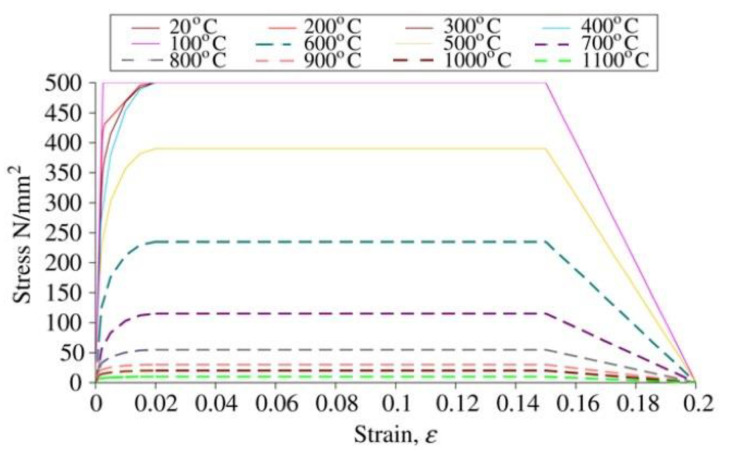
Stress-strain relationship at elevated temperature for structural steel, EC3 British Standards Institution.

**Figure 7 materials-15-01402-f007:**
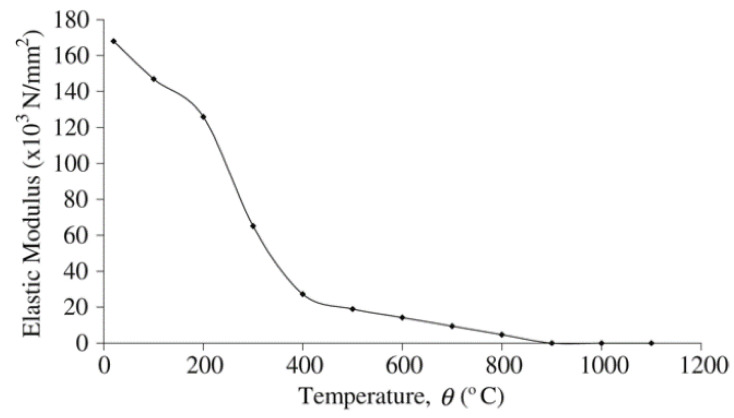
Modulus of elasticity of structural steel at elevated temperatures, EC3 British Standards Institution.

**Figure 8 materials-15-01402-f008:**
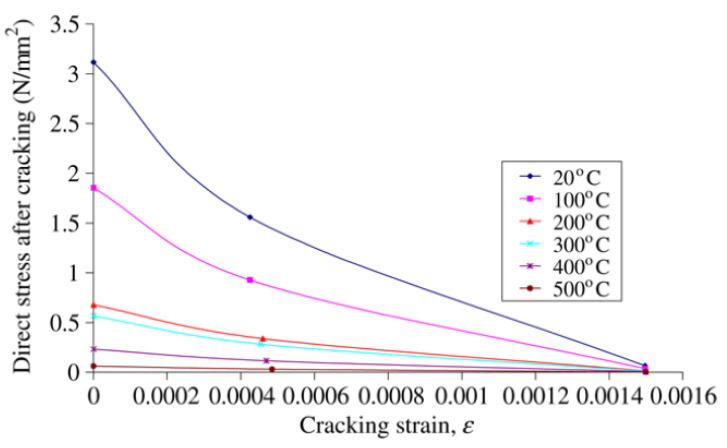
Tensile stress-strain relationship at elevated temperature for concrete, EC2 British Standards Institution.

**Figure 9 materials-15-01402-f009:**
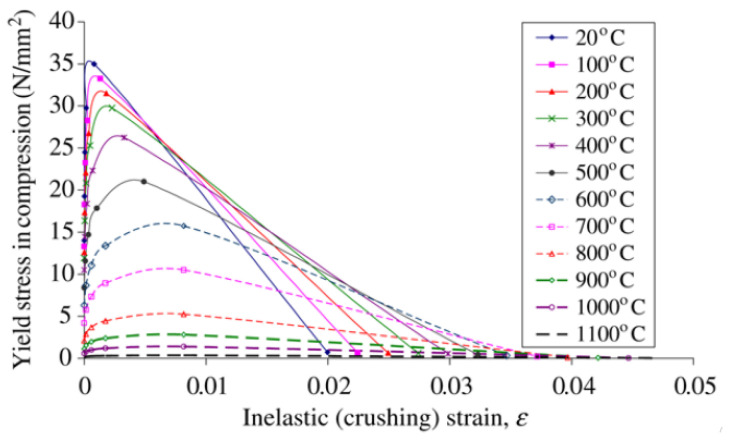
Compressive stress–strain relationship at elevated temperature for concrete, EC2 British Standards Institution.

**Figure 10 materials-15-01402-f010:**

Step-by-step deformation of the model.

**Figure 11 materials-15-01402-f011:**
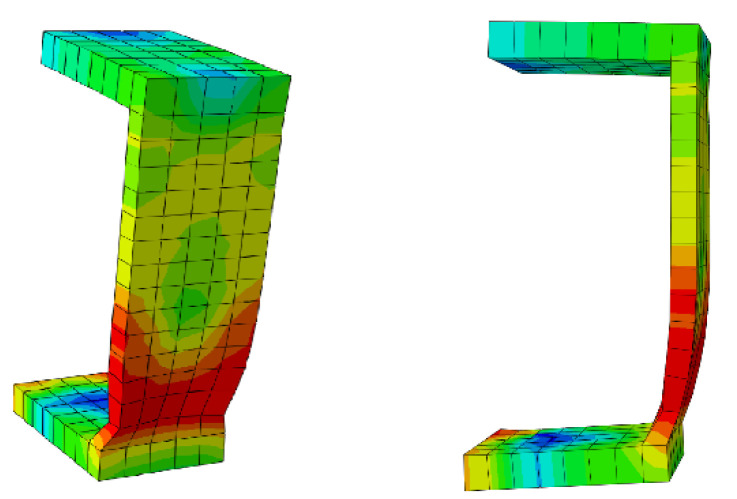
Typical channel connector failure at elevated temperature.

**Figure 12 materials-15-01402-f012:**
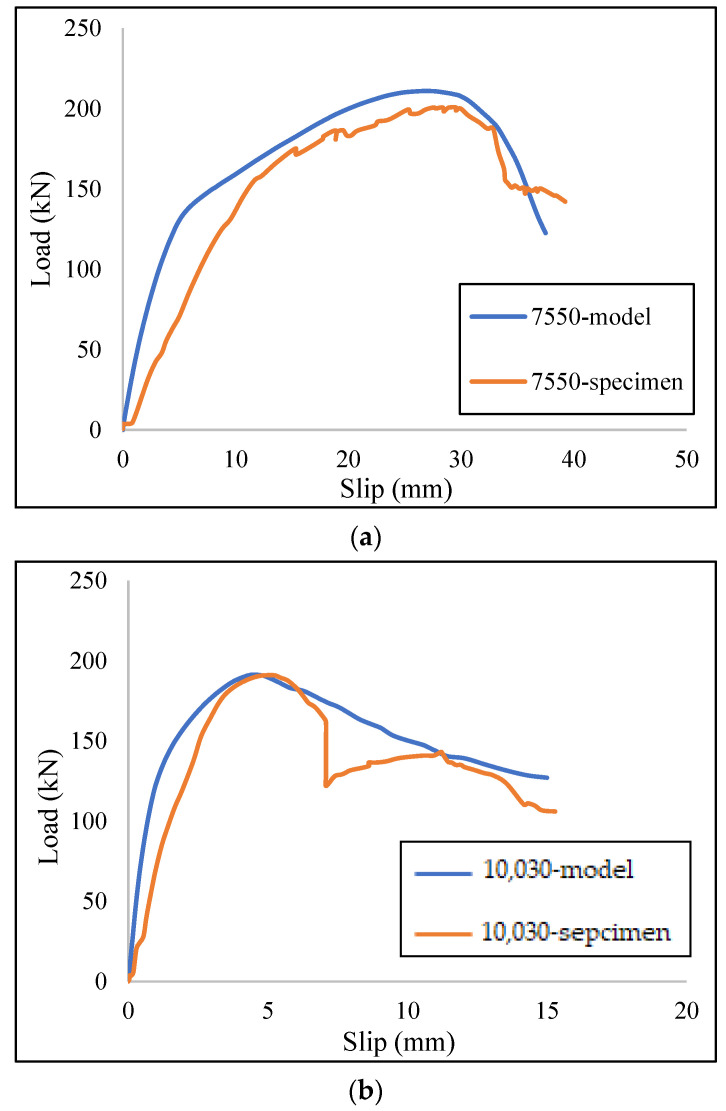
Comparative load–slip curves for (**a**) C7550 type connector at 550 °C, and (**b**) C10030 type connector at ambient temperature.

**Figure 13 materials-15-01402-f013:**
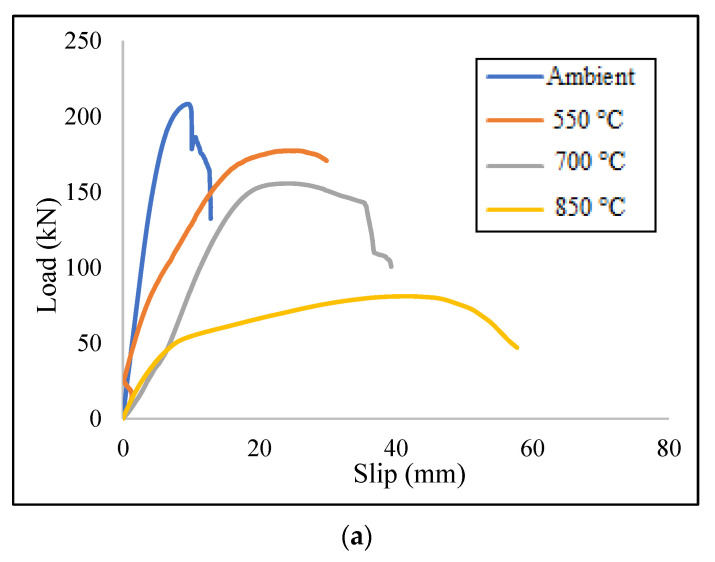
Load-slip curves of the FE models for (**a**) model with C7530, (**b**) model with C7550, (**c**) model with C10030 and (**d**) model with C10050 shear connector.

**Figure 14 materials-15-01402-f014:**
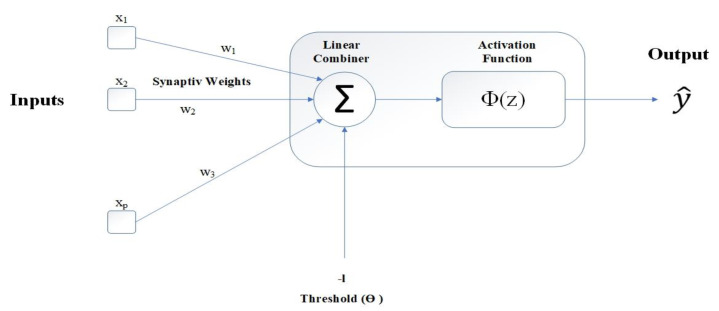
Schematic representation of MLP neuron.

**Figure 15 materials-15-01402-f015:**
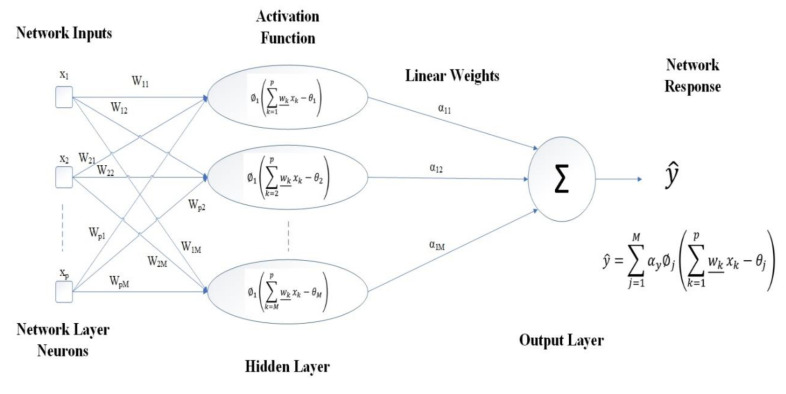
Flowchart of typical single line hidden layer MLP for identifying a problem.

**Figure 16 materials-15-01402-f016:**
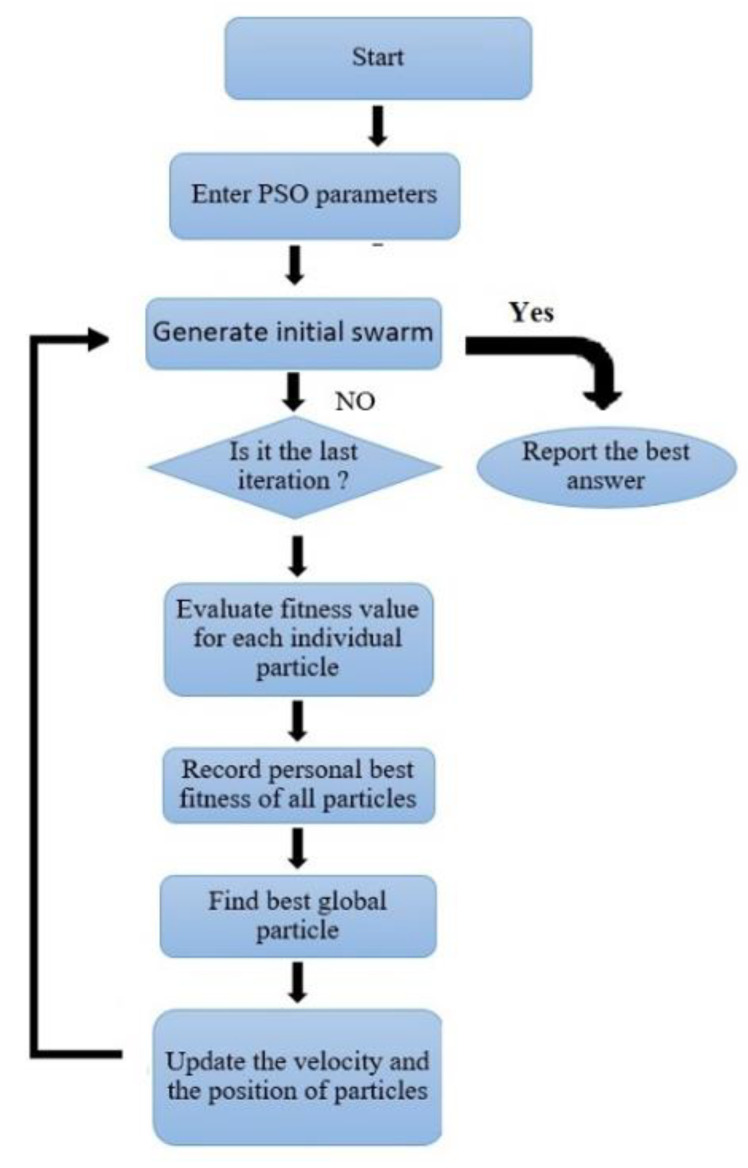
PSO sequential flowchart [57].

**Figure 17 materials-15-01402-f017:**
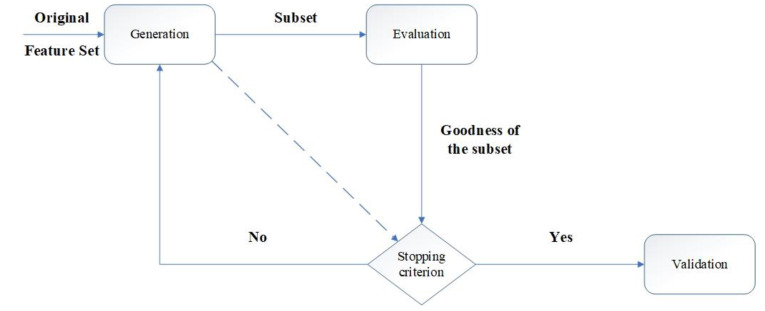
FS method flowchart.

**Figure 18 materials-15-01402-f018:**
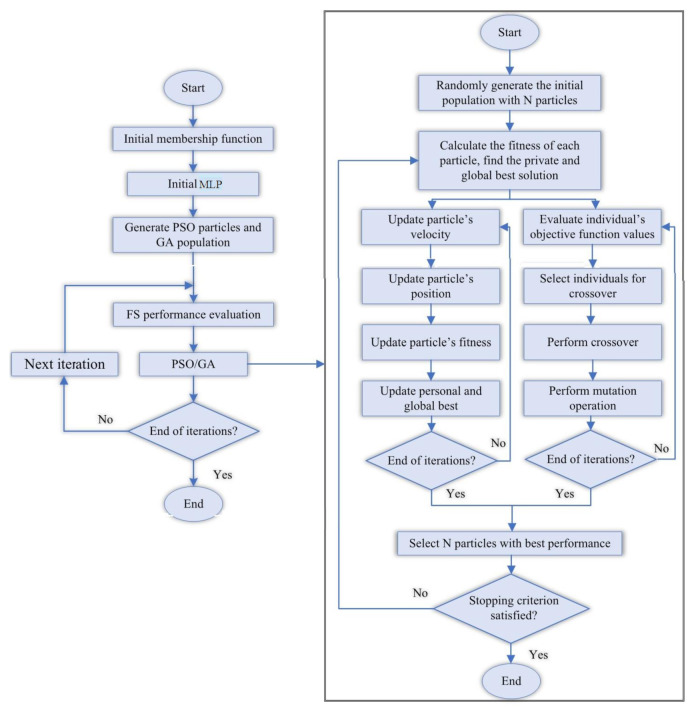
Flowchart of the sequential combination of hybrid MPF algorithm.

**Figure 19 materials-15-01402-f019:**
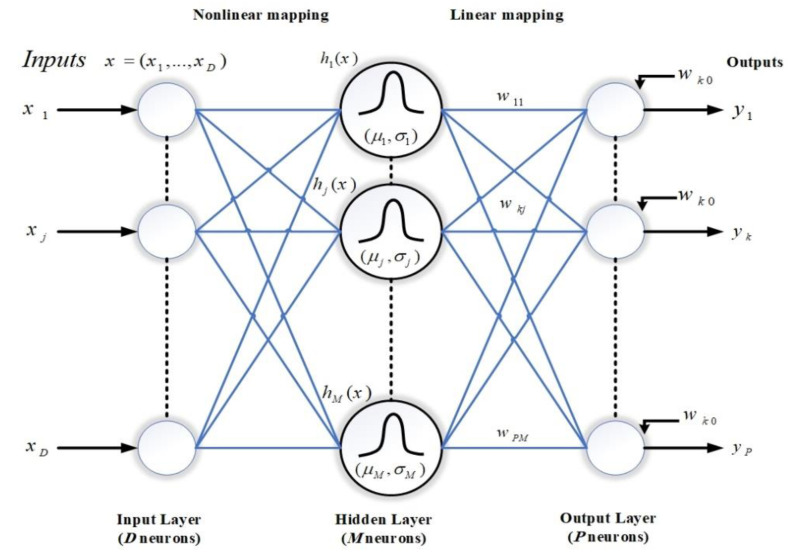
The architecture of an RBF network.

**Figure 20 materials-15-01402-f020:**
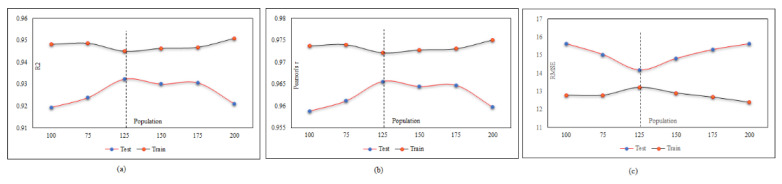
MPF adjustment based on the population number: (**a**) effect of the population number on (*R*^2^), (**b**) effect of the population number on (r), and (**c**) effect of the population number on (*RMSE*).

**Figure 21 materials-15-01402-f021:**
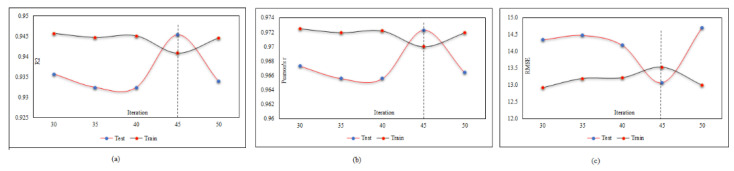
MPF adjustment based on the iteration number: (**a**) effect of the iteration number on (*R*^2^), (**b**) effect of the iteration number on (r), and (**c**) effect of the iteration number on (*RMSE*).

**Figure 22 materials-15-01402-f022:**
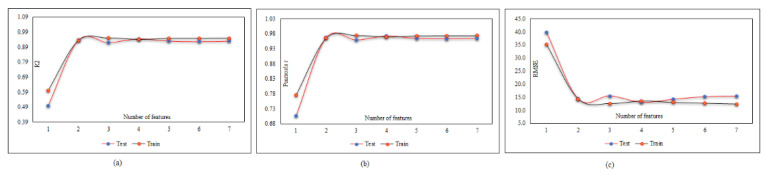
MPF adjustment based on inputs combination: (**a**) effect of inputs combination on *R*^2^, (**b**) effect of inputs combination on r, and (**c**) effect of inputs combination on *RMSE*.

**Figure 23 materials-15-01402-f023:**
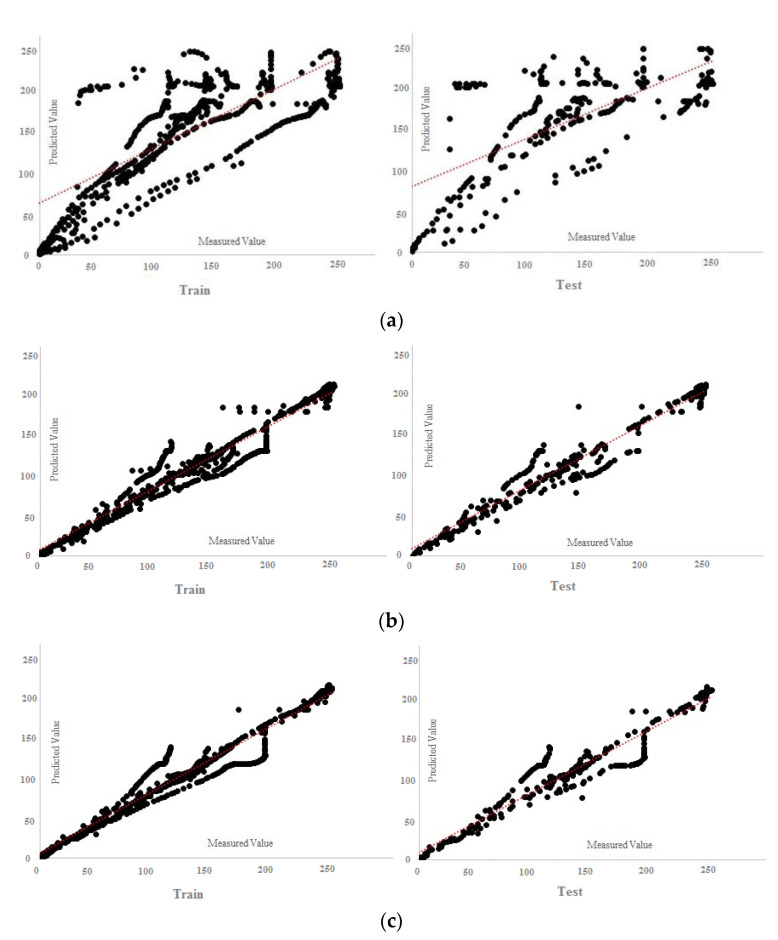
MPF regression charts: (**a**) one input, (**b**) two inputs, (**c**) three inputs.

**Figure 24 materials-15-01402-f024:**
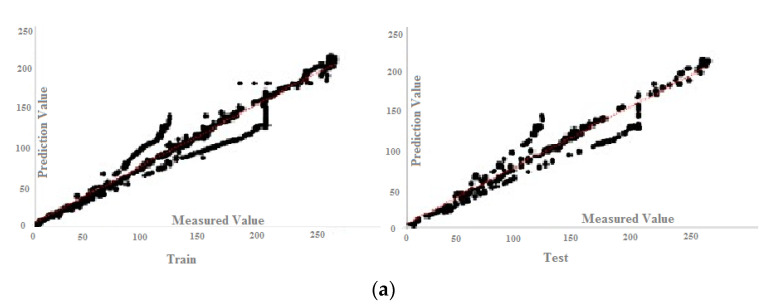
MPF regression charts: (**a**) four inputs, (**b**) five inputs, (**c**) six inputs, (**d**) seven inputs.

**Figure 25 materials-15-01402-f025:**
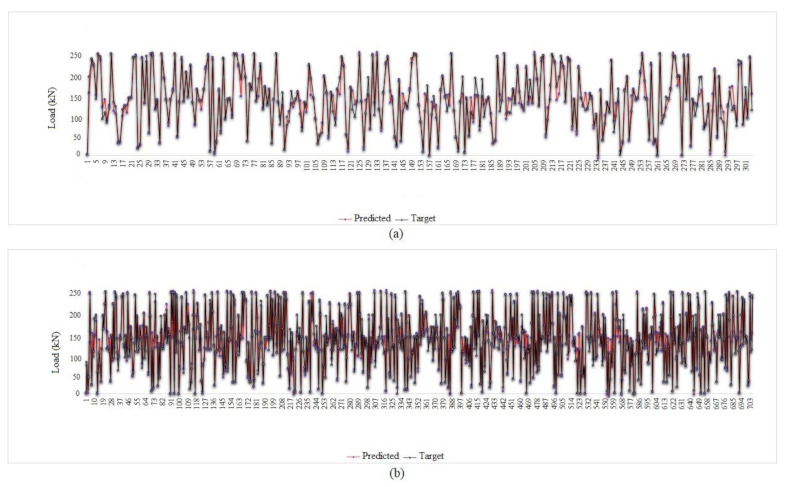
Tolerance diagram with respect to the MPF (four inputs) prediction: (**a**) test phase, (**b**) train phase.

**Figure 26 materials-15-01402-f026:**
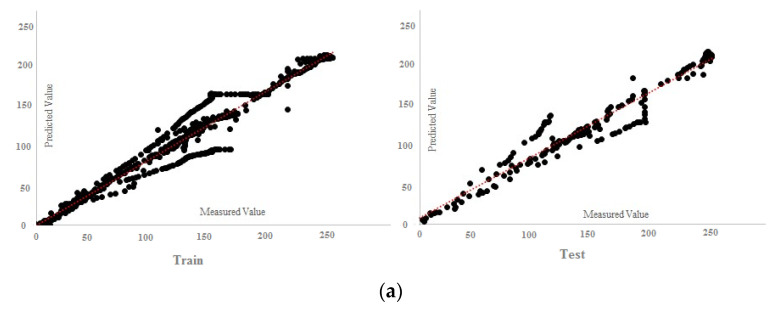
ELM and RBF regression charts: (**a**) ELM (four inputs), (**b**) ELM (seven inputs), (**c**) RBF (seven inputs) and (**d**) RBF (four inputs).

**Figure 27 materials-15-01402-f027:**
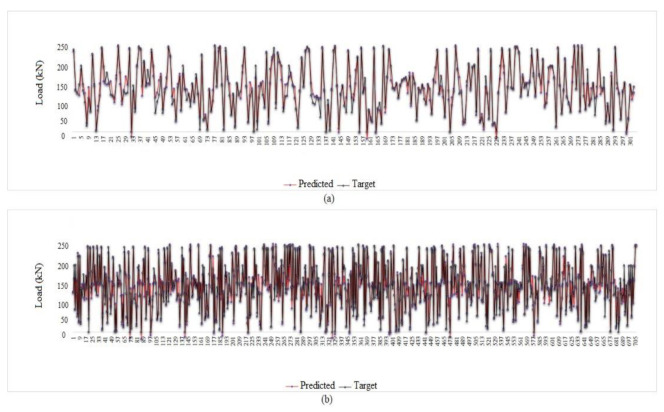
Tolerance diagram with respect to the RBF (seven inputs) prediction: (**a**) test phase, and (**b**) train phase.

**Figure 28 materials-15-01402-f028:**
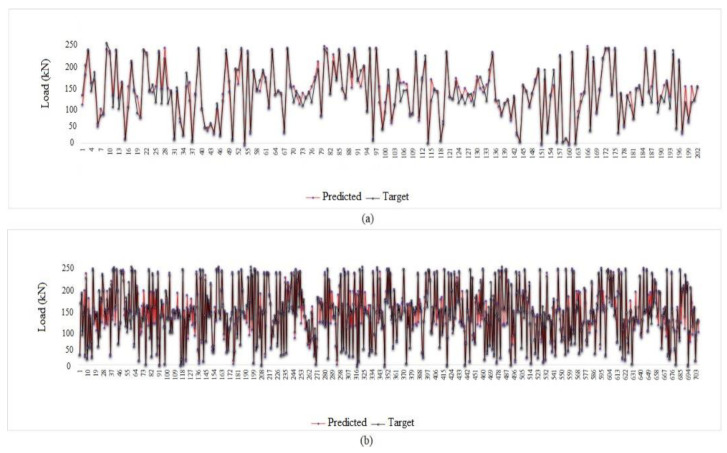
Tolerance diagram with respect to the ELM (seven inputs) prediction: (**a**) test phase, (**b**) train phase.

**Figure 29 materials-15-01402-f029:**
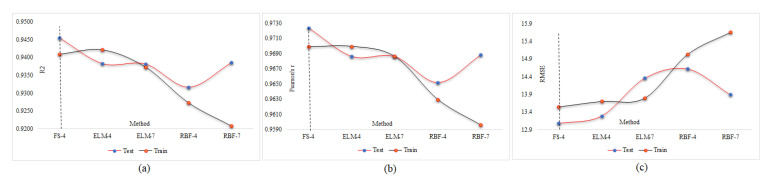
Comparing performance parameters for different methods: (**a**) *R*^2^, (**b**) *r*, and (**c**) *RMSE*.

**Table 1 materials-15-01402-t001:** Simulated channel geometric features.

**Channel Type**	**Geometry (mm)**			**Channel View**
	**Length**	**Width**	**Thickness**	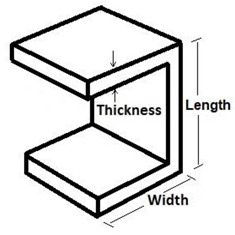
7550	75	50	5.5	
10,030	100	30	7.0	
7530	75	30	5.5	
10,050	100	50	7	

**Table 2 materials-15-01402-t002:** FE analysis results.

Channel Type	Temperature (°C)	Failure Load (kN)	Maximum Slip (mm)
C7530	Ambient	169.24	13.51
	550	144.00	28.17
	700	124.36	38.11
	850	42.10	57.83
C7550	Ambient	260.52	13.17
	550	218.74	36.95
	700	160.19	43.56
	850	109.08	52.21
C10030	Ambient	192.26	16.5
	550	167.19	44.27
	700	135.41	60.04
	850	71.38	78.33
C10050	Ambient	215.03	11.43
	550	168.55	26.04
	700	149.89	49.21
	850	73.56	76.54

**Table 3 materials-15-01402-t003:** Details of the input and output variables.

Inputs and Outputs	Variables	Minimum	Maximum	Mean Value	Standard Deviation
Input 1	Load (kN)	0.006	260.52	105.67	55.869
Input 2	Length(mm)	30	50	41.9	9.823
Input 3	fc(N/mm2)	38.2	82	50.97	17.266
Input 4	Connector-thickness(mm)	5.5	7.0	5.49	0.500
Input 5	profile-thickness(mm)	7.5	8.5	7.99	0.500
Inputs 6	Height(mm)	75	100	87.25	12.504
Input 7	Temperature(C)	25	850	521.56	328.455
Output	Slip (mm)	0.024	78.33	37.44	24.088

**Table 4 materials-15-01402-t004:** The used parameter characteristics for PSO in this study.

FIS Clusters	Population Size	Iterations	Inertia Weight	Damping Ratio	Learning Coefficient	
					Personal	Global
10	125	45	1	0.99	1	2

**Table 5 materials-15-01402-t005:** Parameter characteristics used for MLP in this study.

Hidden Layers	Training Function
10	Levenberg–Marquardt Backpropagation (LMBP)

**Table 6 materials-15-01402-t006:** Parameter characteristics used for RBF in this study.

Mean Squared Error Goal	Spread of Radial Basis Functions	Maximum Number of Neurons
0.02	10	40

**Table 7 materials-15-01402-t007:** Parameter characteristics used for FS in this study.

Number of Runs	Number of Functions (nf)
3	4

**Table 8 materials-15-01402-t008:** The calculated accuracy criteria for the performance of the implemented models (iterations = 40).

Population	Network Result					
	Testing Phase			Training Phase		
	*RMSE*	*r*	$R^{2}$	*RMSE*	*r*	$R^{2}$
100	15.621	0.959	0.919	12.785	0.974	0.948
75	15.028	0.961	0.924	12.787	0.974	0.949
125	14.182	0.966	0.932	13.217	0.972	0.945
150	14.812	0.964	0.930	12.909	0.973	0.946
175	15.300	0.965	0.931	12.682	0.973	0.947
200	15.619	0.960	0.921	12.403	0.975	0.951

**Table 9 materials-15-01402-t009:** The calculated accuracy criteria for the performance of the implemented models (Population = 125).

Iteration	Network Result					
	Testing Phase			Training Phase		
	*RMSE*	*r*	$R^{2}$	*RMSE*	*r*	$R^{2}$
30	14.347	0.967	0.936	12.917	0.972	0.946
35	14.478	0.966	0.932	13.193	0.972	0.945
40	14.182	0.966	0.932	13.217	0.972	0.945
45	13.072	0.972	0.945	13.533	0.970	0.941
50	14.708	0.966	0.934	12.991	0.972	0.945
30	14.347	0.967	0.936	12.917	0.972	0.946

**Table 10 materials-15-01402-t010:** The calculated accuracy criteria for the performance of the implemented models for different input numbers.

Combination Number	Network Result					
	Testing Phase			Training Phase		
	*RMSE*	*r*	$R^{2}$	*RMSE*	*r*	$R^{2}$
1	39.825	0.706	0.498	35.160	0.775	0.601
2	14.408	0.967	0.936	14.344	0.966	0.933
3	15.490	0.959	0.921	12.570	0.975	0.950
4	13.072	0.972	0.945	13.533	0.970	0.941
5	14.199	0.964	0.930	13.004	0.973	0.947
6	15.180	0.963	0.927	12.665	0.974	0.948
7	15.210	0.961	0.929	12.415	0.972	0.948

**Table 11 materials-15-01402-t011:** Most effective inputs based on feature selection.

Feature	Number of Combination						
	1	2	3	4	5	6	7
Load (kN)	X	X	X	X	X	X	X
Length(mm)					X	X	X
fc(N/mm^2^)				X	X	X	X
connector-thickness(mm)						X	X
profile-thickness(mm)				X			X
Height(mm)			X		X	X	X
Temperature(°C)		X	X	X	X	X	X

**Table 12 materials-15-01402-t012:** The calculated accuracy criteria for the performance of the implemented models for different neural networks.

Method	Network Result					
	Testing Phase			Training Phase		
	*RMSE*	*r*	$R^{2}$	*RMSE*	*r*	$R^{2}$
FS-4 inputs	13.072	0.972	0.945	13.541	0.970	0.941
ELM-4 inputs	13.286	0.969	0.938	13.699	0.970	0.942
ELM-7 inputs	14.356	0.969	0.938	13.791	0.969	0.937
RBF-4 inputs	14.621	0.965	0.932	15.029	0.963	0.927
RBF-7 inputs	13.884	0.969	0.939	15.656	0.960	0.921

## Data Availability

Data sharing is not applicable to this article.

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
