# Peer review of "Evaluation of the Performance of a Composite Profile at Elevated Temperatures Using Finite Element and Hybrid Artificial Intelligence Techniques"

_materials, 2022, doi:10.3390/ma15041402_

Round 1
Reviewer 1 Report
Review report on paper materials-1583738-peer-review-v1, entitled “Evaluation of the performance of a composite profile at elevated temperatures using finite element and hybrid artificial intelligence techniques”
This paper suggests using a novel hybrid intelligence algorithm MPF(MLP-PSO-FS) to Investigate the structural behaviour of a composite model with a channel connector at elevated temperatures. It is well known that even though artificial intelligence algorithms are abundant, existing algorithms still cannot match exactly and apply to every research objective. In this paper, a new algorithm is proposed for a specific composite model, and it is proved by theoretical calculation that this algorithm has advantages over traditional AI algorithms ELM and RBF. This is the main contribution of this paper. The data required by the algorithm is calculated by the FEM simulation model. This reviewer believes that this work puts forward a new algorithm suitable for theoretical analysis so that it deserves to be published in MATERIALS. However, the following minor issues are needed to consider:
- There are too many keywords, please consider reducing them.
- There lack some background on composite materials in the introduction.
[1] Optimal design of broadband quasi-perfect sound absorption of composite hybrid porous metamaterial using TLBO algorithm
[2] Design and study of a hybrid composite structure that improves electromagnetic shielding and sound absorption simultaneously
[3] Annular acoustic black holes to reduce sound radiation from cylindrical shells
[4] Transmission loss of plates with multiple embedded acoustic black holes using statistical modal energy distribution analysis
[5] Noise reduction via three types of acoustic black holes
- There are too many blanks in Fig. 4 (also in Fig. 5c-c and Fig. 16), the figure should be cut properly and placed in one row. Moreover, the caption in this figure should be on the same page as the figure. This is just a format issue and it could be improved in publication.
- The legend in Fig. 6 does not match the lines in the figure.
- Equations: Eqs. 4-6 are badly edited. The equation numbers need to be (1)-(6). Also, variables should be italic.
- There are too many preparations in Section 5, they should be suppressed and the focus should be placed on the main contribution, e.g., shear connector and the comparisons with other algorithms.

Author Response
We would like to appreciate the reviewer for the valuable comments and delicate review. We have done our best to address the comments. Please kindly see the attachment.

Reviewer 2 Report
This paper is original and interesting. It is clearly and well written. The methodology is adequately described, while the results and conclusions are clearly presented. This paper can be accepted for publication with minor changes.
Below are a couple of suggestions that I think would improve the paper.
In Figure 6, the legend needs to be corrected because it is not noticeable which curve is full and which is dashed.
In Figure 13, the temperature units need to be repaired. In expressions (1) and (2) the markings are not consistent with the explanations below.
The notations of equations (4), (5) and (6) should be equated with (1) and (2).
Author Response
We are grateful to the reviewer for her/his attention and valuable comments. We have done our best to address the comments. Please kindly see the attachment.

Reviewer 3 Report
The hybrid intelligence algorithm that uses artificial neural networks is performed on derived data from the FE study. They tried to fit their data listed in Table 6 with their proposed algorithm MLP with PSO algorithm. However, the reviewer does not recommend this paper for publication due to the following reasons.
(2) What kinds of software were used? If you employ any in-house code or specific software, you have to show the name of the software.
(3) Also, the hyper-parameter tuning procedure was missing according to the number of the neurons and the number of layers, the kind of solvers, increasing the performance dramatically among the given architecture of MLP. Thus, In the current state, it is hard to say that their contribution was impressive.
(3) Also, Fig. 24 and 25 do not show a linear regression between prediction and validation results. It might be due to noisy data but has to be fixed
Author Response
We want to thank the reviewer for her/his precise attention and invaluable comments. We have done our best to address the comments. Please kindly see the attachment.

Reviewer 4 Report
The authors consider an innovative method for characterization of mechanical properties of a composite profile used in civil engineering. The method is based on artificial intelligence techniques and it is aimed at predicting the material response for elevated temperatures. The subject of the study is within the scope of MDPI Materials.
The authors have proven their practical skills and knowledge as far as the use of state-of-the-art AI techniques is concerned. Their approach is based on the FEM computations carried out in the ABACUS environment, these are applied to collect numerical data used in subsequent post-processing. The next stage uses multi-layer neural network combined with particle swarm optimization and feature selection for prediction of mechanical quantities like failure load, maximum slip etc. for the considered component. The predictions of the proposed hybrid algorithm are compared against those from radial basis function neural networks (RBF) and extreme learning machine (ELM).
The paper might be potentially useful for the designers of complicated components used e.g. in civil engineering. However before the manuscript is accepted, it should be supplemented with more details concerning the FEM method used (the number of finite elements used, their type, the computation time and the machine used).
Author Response
We gratefully appreciate the reviewer comments that were invaluable and helpful for the paper. We did our best to address the comments. Please kindly see the attachment.

Round 2
Reviewer 3 Report
It is hard to agree with this data and understand this data in Figs. 24 and 26. Prediction values and Measure values show one-to-one correspondence. Also, in MATLAB, PYTORCH, there are many excellent MLP algorithms. It can be realized easily. However, there was no comparative study that demonstrated the authors' algorithm better. Thus, I can't entirely agree that this work has much novelty.